# CODING TRIANGLE: HOW DOES LARGE LANGUAGE MODEL UNDERSTAND CODE?

## ABSTRACT

Large language models (LLMs) have made remarkable progress in code generation. However, the evaluation of their actual programming capabilities remains largely limited to solving standard coding problems, and a full understanding is under exploration. We propose **Coding Triangle**, a systematic approach for evaluating LLMs across three core dimensions: code editing, code implementation, and test case generation. Through comprehensive experiments on competitive programming benchmarks, we evaluate the model performance across these dimensions and uncover both self-consistency and self-inconsistency within the model's own cognition. Self-consistency often results in solutions that lack the diversity and robustness seen in human programmers, leading to a significant distribution shift between model cognition and human submissions. Our analysis of interactions between different dimensions reveals that self-inconsistency also exists, which may enable self-reflection and self-improvement and provide a potential direction for developing more powerful coding models.

## 1 INTRODUCTION

Recent advances in large language models (LLMs) with rapid development in model design and data scaling (Achiam et al., 2023; OpenAI, 2025b; Dubey et al., 2024; Jiang et al., 2023; Yang et al., 2024a; Team, 2025a; Anthropic, 2024; OpenAI, 2024) have driven remarkable progress on code generation benchmarks (Austin et al., 2021b; Cassano et al., 2022; Liu et al., 2024b; Li et al., 2024; Guo et al., 2024; Wu et al., 2024). For example, DeepSeek-V3 (Liu et al., 2024a) achieves a score of 65.2 on HumanEval (Guo et al., 2025; Chen et al., 2021b), Qwen3 attains 65.7 on LiveCodeBench (Team, 2025a; Jain et al., 2024), and o3 reaches a Codeforces Elo rating of 2724 (OpenAI, 2025b; El-Kishky et al., 2025), demonstrating impressive coding abilities of modern foundation models.

As large language models continue to advance in coding capabilities, there is increasing concern that current coding benchmarks may not accurately or comprehensively evaluate the true performance of LLMs. In this work, we revisit a foundational question: **How should the coding capability of LLM be defined?** Taking inspiration from the typical workflow of human developers when solving coding problems, we note that they usually follow a structured pipeline that includes: (1) analyzing the problem, (2) designing a preliminary solution strategy, (3) implementing the code, (4) performing manual testing, and (5) iteratively refining the solution until it passes all test cases.

Motivated by this, we investigate the coding capabilities of LLMs through three interconnected dimensions: **Editorial**, **Code**, and **Cases**. We develop such a three-dimensional analysis framework called **Coding Triangle** to study and better understand the coding behaviors of LLMs. Our goal is to understand how LLMs fundamentally interpret coding problems within each dimension and to explore the interactions across them. For example: *(a) Does performance across dimensions exhibit consistency? (b) Does a model's code generation benefit from its natural language problem breakdowns? (c) Does the generated code reliably pass self-generated test cases? (d) To what extent do these test cases adequately reflect the reasoning outlined in the editorial?* Based on our Coding Triangle framework, we conduct extensive experiments on 200 problems collected from AtCoder and evaluate various LLMs including general models, coding models, and reasoning models. By analyzing their capabilities and interactions among different dimensions, we gain deeper insights into how LLMs truly comprehend and tackle coding tasks with several surprising and interesting findings.

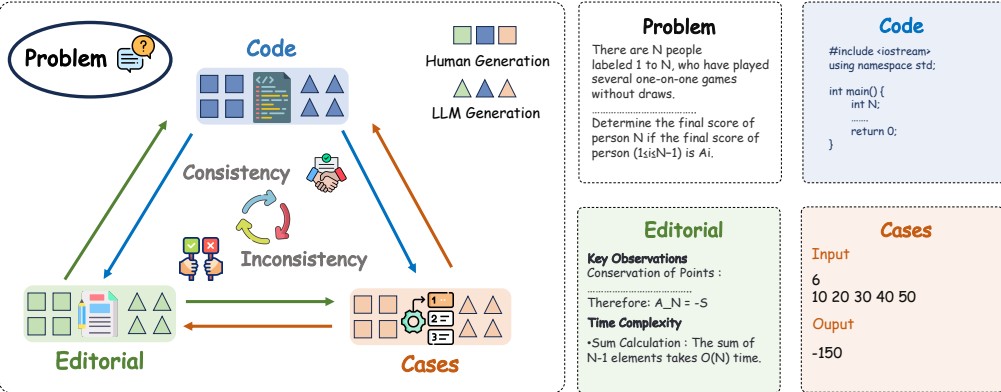

Figure 1: **The framework of Coding Triangle.** Editorial, code, and cases form the three fundamental vertices of the triangle, with each vertex being sampled either from human or from models. These dimensions are interconnected, influencing one another, and their relationships form the six directed edges of the triangle.

Our results reveal widespread **self-consistency** across three dimensions in various LLMs. This self-consistency often constrains LLM reasoning from exploration, resulting in converge on narrow patterns and similar mistakes, particularly in edge cases or implementation-specific details. As a result, a significant distribution shift emerges between LLM predictions and human solutions. Notably, ensembling multiple models can effectively mitigate these cognitive biases, promoting greater diversity and robustness in performance. Conversely, we also observe **self-inconsistency** across these dimensions, indicating that the corresponding capabilities are not always well-aligned. For instance, an LLM may accurately identify its own failed solutions and generate test cases that clearly differentiate between correct and incorrect outputs. These issues persist for both non-reasoning and reasoning models, indicating that extended reasoning stage is insufficient to fully resolve self-inconsistency. These findings point to the potential for self-reflection and self-improvement by iteratively aligning these dimensions. To summarize, this paper makes the following contributions.

- We propose a framework called **Coding Triangle** to systematically examine the internal knowledge of LLMs in programming, enabling a comprehensive evaluation of their coding abilities.

- We estimate the performance across different dimensions and investigate the distribution shift between the LLM predictions and actual solutions from humans, finding that incorporating human knowledge can substantially improve performance.

- By analyzing the self-consistency and self-inconsistency inside LLMs, we observe that their strengths and weaknesses vary across the three dimensions in Coding Triangle, demonstrating the advantages of model mixtures and the potential for self-reflection and self-improvement.

## 2 OVERVIEW

In this section, we introduce the three fundamental dimensions of the **Coding Triangle** and define how to evaluate the capabilities of LLMs across these dimensions. We then proceed with our analysis in Section 3, where we examine distributional shifts relative to human coding behaviors, and further discuss the interactions among the three dimensions in Section 4. Our experiments cover general foundation models, code-oriented models, and reasoning-enhanced models[1], and we use AtCoder[2] problems as our evaluation benchmark.

### 2.1 CODING TRIANGLE

In this study, we systematically examine the comprehensive understanding of LLMs when addressing competitive programming challenges. Motivated by the way humans solve problems step by step

---

[1] We refer to DeepSeek-V3 (Liu et al., 2024a), Qwen2.5-72B-Instruct (Yang et al., 2024b), Qwen2.5-Coder-32B-Instruct (Hui et al., 2024), and QwQ-32B (Team, 2025b) as DS-V3, 72B, Coder, and QWQ, respectively.

[2] Problems A–F from AtCoder Beginner Contests (ABC) 175–374.

through analyzing, coding, and manually testing, we formally decompose coding ability into three interconnected perspectives, and establish the overall framework of the **Coding Triangle**:

- *Editorial* captures how LLM interprets and analyzes the problem in natural language, providing the most accessible explanation for human readers.

- *Code* reflects the programming logic and ability of algorithm implementation, serving as the machine-executable counterpart to the human-readable editorial.

- *Cases* indicate the depth of understanding in terms of validation criteria, including edge scenarios and boundary conditions.

Intuitively, these three dimensions create a comprehensive system that captures all aspects of a coding problem, from interpretation to execution and validation. We then introduce evaluation metrics for each dimension, allowing us to quantify its strengths and weaknesses in a structured way.

## 2.2 EVALUATION METRIC

**Editorial.** We adopt an LLM-as-a-judge approach to evaluate the quality of model-generated editorials. Given a model-generated editorial $E_{\mathrm{model}}$ and a ground-truth editorial $E_{\mathrm{gt}}$, we employ o3-mini as the judging model to compare them and assign a correctness score. Let $N$ denote the total number of problems. The overall editorial score $S_{\mathrm{edi}}$ is then defined as:

$$S_{\mathrm{edi}} = \frac{1}{N} \sum_{i=1}^{N} \mathrm{LLM}(E_{\mathrm{model}}^{(i)}, E_{\mathrm{gt}}^{(i)}), \tag{1}$$

where $\mathrm{LLM}(\cdot, \cdot)$ denotes the correctness score predicted by the judge.

**Code.** During the evaluation, the model is prompted to generate a solution for each problem, which is subsequently validated against all the test cases. For each problem $i$, let $\mathcal{T}_i$ denote the set of the test cases, $\mathcal{J}$ the judge function, and $s_i$ the solution generated by the model. Let $N$ be the total number of problems. We then count the number of problems for which the solution passes all test cases: $N_{\mathrm{code}} = |\{i \mid \forall t \in \mathcal{T}_i, \ \mathcal{J}(s_i, t) = \mathrm{Accepted}\}|$. The code score is defined as Pass@1 accuracy:

$$S_{\mathrm{code}} = \frac{N_{\mathrm{code}}}{N}. \tag{2}$$

**Cases.** It is observed that official cases for some problems are not sufficiently comprehensive and may fail to cover all edge scenarios, potentially leading to incorrect solutions being accepted. Therefore, we focus on those errors that can be identified by the official cases and evaluate model-generated cases with only incorrect solutions. Furthermore, directly generating input-output pairs (Chen et al., 2021a; Liu et al., 2023) is often insufficient, as most generated cases are incorrect and will be filtered out. Instead, we prompt the model to generate inputs and use the official solution to produce the outputs. For each problem $i$, let $\mathcal{H}_i^{\mathrm{wrong}}$ denote the set of all human submissions that are incorrect, and $\mathcal{J}(h, \mathcal{C}_i)$ as the judge results of submission $h$ on the model-generated cases $\mathcal{C}_i$. We consider the cases set to be correct if the judge results are consistent with the official evaluation and all wrong solutions are identified as incorrect. Let $N_{\mathrm{case}}$ to be the number of problems for which the model-generated cases can effectively distinguish all the edge cases, we have $N_{\mathrm{case}} = |\{i \mid \forall h \in \mathcal{H}_i^{\mathrm{wrong}}, \ \mathcal{J}(h, \mathcal{C}_i) = \mathcal{J}(h, \mathcal{T}_i)\}|$, where $\mathcal{T}_i$ is the set of ground-truth test cases for problem $i$. The case score is then defined as

$$S_{\mathrm{case}} = \frac{N_{\mathrm{case}}}{N}. \tag{3}$$

## 3 ABILITY ANALYSIS AND DISTRIBUTION SHIFT

With the evaluation metrics established above, we analyze the capabilities of LLMs across the three dimensions in this section. Furthermore, we explore the distribution shift between model cognition and human submissions, and demonstrate the robustness introduced by model mixture.

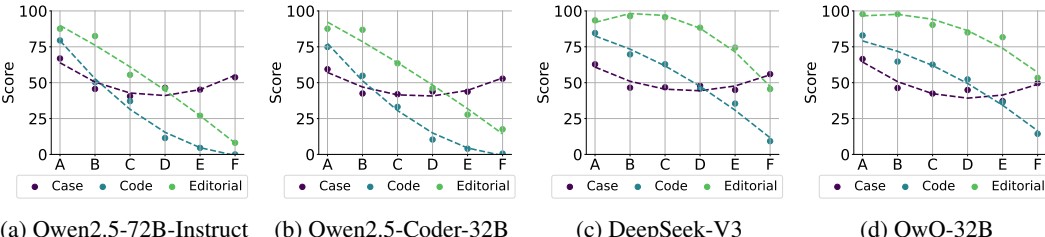

(a) Qwen2.5-72B-Instruct     (b) Qwen2.5-Coder-32B     (c) DeepSeek-V3     (d) QwQ-32B

Figure 2: **Ability analysis across different dimensions.** (1) Both the editorial and code scores decline as difficulty increases, and a notable performance gap emerges between them. (2) In contrast, the case score exhibits a different trend, maintaining excellent performance even on difficult problems.

### 3.1 ABILITY ANALYSIS

We present the performance across the three dimensions in Figure 2 to illustrate their interrelationships. It can be observed that models with strong coding capabilities tend to perform well in both the editorial and code dimensions, suggesting a consistency between these two aspects. As problem difficulty increases, performance in both dimensions declines. Notably, the editorial score consistently exceeds the code score, leading to a performance gap ranging from 5% to 20% when models attempt to transform their problem analysis into executable code. This trend is evident across all models, including reasoning models like QwQ-32B, where correct reasoning does not always yield correct solutions. Furthermore, this gap is most significant on medium-difficulty problems, suggesting the conflict between analyzing and implementation in this level.

In contrast, the case score follows a different trend and does not decrease monotonically with increasing problem difficulty. Surprisingly, we find that on the most difficult problems, the case score can even surpass both the editorial and code scores. Interestingly, performance drops on medium-difficulty problems (C and D) which we attribute to the increased complexity and coverage of edge cases at this level. For harder problems such as E and F, the primary challenge shifts toward algorithm selection and techniques implementation, rather than handling numerous edge conditions, leading to a rebound in case accuracy. These observations suggest a notable inconsistency that the ability to generate test cases does not fully align with editorial analysis or code generation.

### 3.2 DISTRIBUTION SHIFT

In practice, LLMs are capable of performing problem analysis, code generation, and test case generation to form a self-consistent system. However, this system is actually limited to its own cognition, and has a significant distribution shift from human solutions. In the following section, we investigate how this self-cognition leads to such distribution shifts in code and test cases, as these two dimensions can be used to cross-validate each other.

**Distribution Shift on Code.** To analyze the distribution shift between model and human solutions, we first construct a performance matrix $P_{\text{code}} \in \mathbb{R}^{m \times n}$ for each problem, where $m$ is the total number of solutions (from both models and humans), and $n$ is the number of test cases. Each entry $P_{ij}$ is assigned a value of $1$ if solution $i$ passes test case $j$, and $-1$ otherwise. Each row of $P_{\text{code}}$ represents the performance vector of a specific solution and we exclude solutions that pass all test cases for error analysis. We compute the normalized cosine similarity between every pair of solutions: $\text{sim}(i, j) = \frac{P_i \cdot P_j}{|P_i||P_j|}, \quad \forall i, j \in 1, \ldots, m$. As visualized in Figure 3a, our results reveal that model-generated solutions are highly similar to each other, with most pairs exhibiting a similarity score above $0.8$. This suggests that model tends to make similar mistakes and generating multiple roll-outs does not prevent the inherent reasoning patterns. Notably, solutions produced by the same model show even higher internal cognition, while human-submitted solutions are much more diverse, displaying lower similarity scores and a wider variety of errors.

To further quantify the diversity of solutions, we construct a set of unique performance vectors for each problem and compute the average size of these sets, as shown in Figure 3c. Our results show that human submissions exhibit significantly greater diversity than those generated by models, indicating a wider variety of errors in human attempts. As problem difficulty increases, both model and human

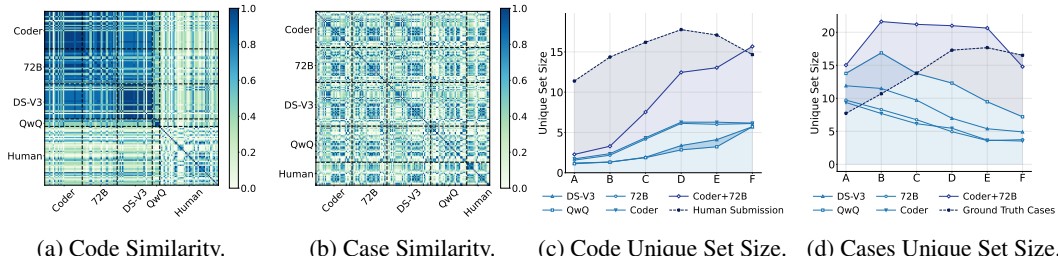

(a) Code Similarity.  (b) Case Similarity.  (c) Code Unique Set Size.  (d) Cases Unique Set Size.

Figure 3: Similarity analysis and unique set size for error analysis of codes and cases.

solutions show a rise in error diversity, reflecting the growing complexity of the tasks. Notably, aggregating solutions from multiple models leads to a broader range of unique behaviors than any single model, suggesting that different models tend to make different types of errors. By integrating solutions from various models, we can uncover more boundary conditions, and a comprehensive analysis of these model-specific errors may further improve overall robustness and performance.

**Distribution Shift on Case.** We also perform a similarity analysis on test cases by constructing a performance matrix $P_{case}$ for each problem, where rows correspond to human submissions and columns represent model-generated and official test cases. Each column is treated as the performance vector of an individual test case, and we calculate the similarity between every pair of test cases, as illustrated in Figure 3b. Unlike solution vectors, model-generated and ground-truth test cases do not exhibit extremely high similarity overall. However, we observe that test cases generated by the same model, as well as official cases, tend to be more similar within their respective groups. This can be attributed to the fact that both model-generated and ground-truth cases are often created using fixed templates or heuristics, resulting in internal redundancy within cases produced by a single source.

We use the unique set as an analytical tool to evaluate the test case diversity as well. Notably, the unique set size produced by the models exceeds that of the official cases for easier problems. A closer examination of the official test cases in these instances reveals that they are typically short and simple, reflecting the low complexity of the problems. Consequently, model-generated cases are able to capture a broader range of potential errors. For more difficult problems, however, model-generated test cases fail to recognize different error from human solutions and the size of the unique set decreases. Interestingly, we observe that combining test cases from different models results in a significantly larger unique set size, indicating that aggregating test cases from multiple models offers better generalization and broader coverage of possible edge cases and error modes.

## 4 Bridging the Edges in Coding Triangle

With the analytical framework established above, we now investigate the interconnections among the three dimensions, with a particular emphasis on the context of self-cognition within LLMs and how the incorporation of external human knowledge such as ground truth editorials, reference solutions, and test cases affects these interrelationships. To this end, we systematically analyze the six possible directed edges of the triangle, investigating how providing information about one aspect, whether from the model itself or from human sources, influences another aspect (X).

### 4.1 From Editorial to X

*From Editorial to Code: Does LLM benefit from the editorial generated by itself?*

To have a better understanding of the gap between the ability to analyze problems and to implement code in LLM, we inspect the performance when feeding its own editorial to generate code, and study how its own problem understanding affects code implementation. As shown in Figure 4, We observe that **providing self-generated editorials does not significantly enhance coding performance**. This suggests that the stage of problem analysis and code implementation are largely self-consistent. Even for reasoning-oriented models like QwQ, which perform reflection on both the editorial and the generated code, there is no notable performance boost. In some instances, prompting the model to

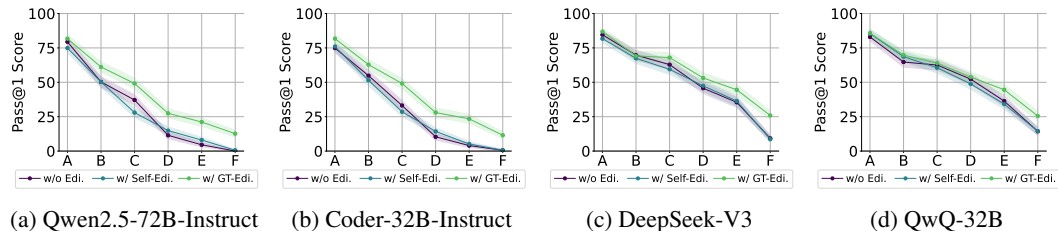

(a) Qwen2.5-72B-Instruct    (b) Coder-32B-Instruct    (c) DeepSeek-V3    (d) QwQ-32B

Figure 4: Pass@1 score with self-generated and ground truth editorials.

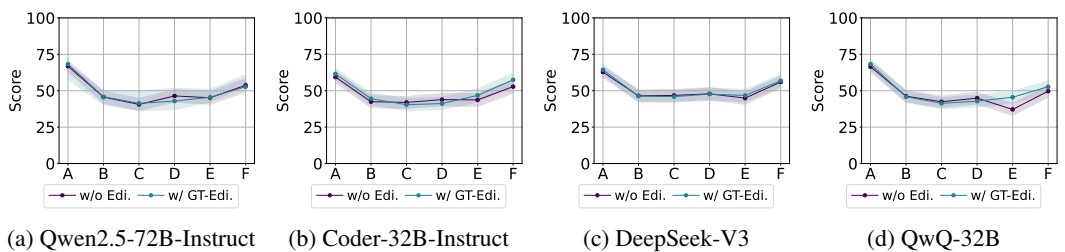

(a) Qwen2.5-72B-Instruct    (b) Coder-32B-Instruct    (c) DeepSeek-V3    (d) QwQ-32B

Figure 5: Case score w.r.t. ground truth editorials.

write an editorial before coding can encourage CoT reasoning and slightly improve the pass rate. However, if the editorial contains flawed analysis, referencing it may actually reduce performance.

In contrast, providing ground truth editorials leads to a much more significant improvement in coding performance. However, even with access to ground truth editorials, models still fail to pass difficult problems, and high success rates are not always achieved, indicating the upper bound of its ability to utilize correct analysis for code generation. Notably, we surprisingly find that **DS-V3 and QwQ display very similar patterns**. For easy problems, neither model gains much from ground truth editorials, suggesting that their internal reasoning and implementation details remain the primary bottlenecks. For harder problems, both models show comparable performance when prompted with ground truth editorials, implying that the main factor influencing the original pass rate is their reasoning and understanding of the problem, rather than their code generation capability.

***From Editorial to Cases: Does case generation benefit from the ground truth editorials?***

We further investigate the impact of editorials on test case generation. As shown in Figure 5, **the ability to generate high-quality test cases does not significantly improve when provided with detailed human-written editorials**. This indicates that the skills involved in creating diverse and comprehensive test cases are different from those required for code generation and are not effectively conveyed through editorials. In particular, test case creation is more closely tied to the implementation details of the solution, while editorials offer only high-level abstractions and are less effective in guiding the model to consider all edge scenarios.

## 4.2 FROM CODE TO X

***From Code to Editorial: Can the LLM Recognize Its Own Mistakes?***

The *editorial* dimension in Coding Triangle serves as a comprehensive reflection of how to understand and analyze the problem. To demonstrate the self-evaluation capability of the LLM, we provide LLM with a candidate solution and ask it to determine whether the code is correct. This setup further allows us to investigate its ability to detect errors in its own output, thereby offering insights into its potential for self-reflection and continuous improvement. We present the results in Figure 6.

When evaluating human-generated solutions, we observe that judge accuracy consistently decreases as problem difficulty increases. This is expected since, as tasks become more complex, the model finds it more challenging to fully understand the problem requirements and to follow the diverse logic present in human-written solutions. However, when the model evaluates its own solutions, we find that judge accuracy increases on the most difficult problems (F). This suggests that **the model**

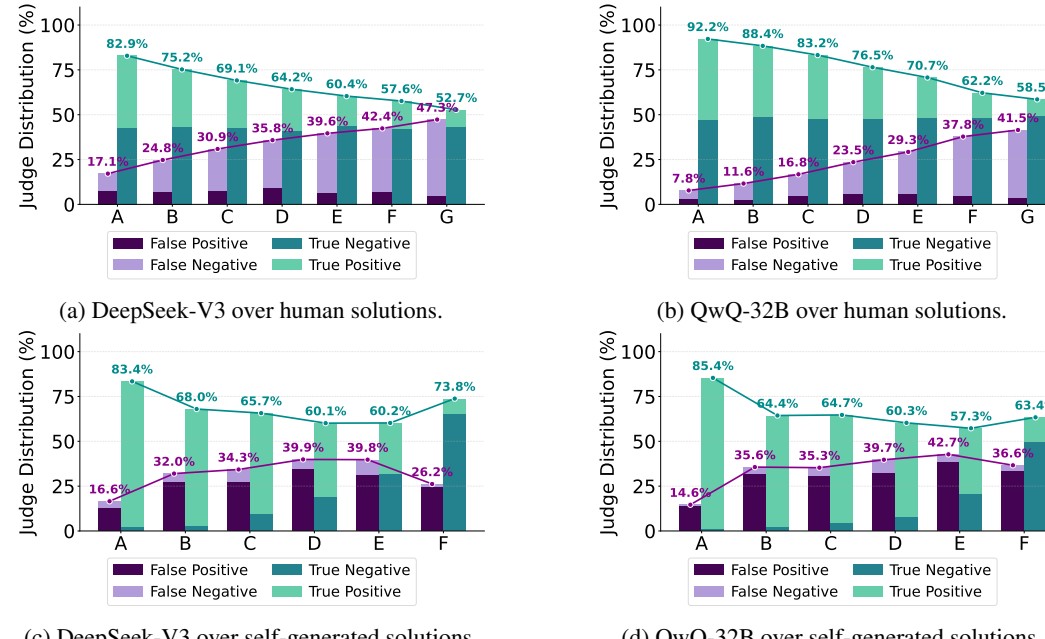

Figure 6: Judge distribution over solutions.

**is still capable of recognizing its own mistakes even when producing incorrect solutions on challenging tasks**. This trend resembles human behavior, where confidence tends to be high on both easy problems and very difficult ones where failure is anticipated. And uncertainty peaks on problems of intermediate difficulty, where errors are harder to detect. Interestingly, similar self-awareness is observed in reasoning models such as QwQ, while the rebound is less pronounced than in DS-V3, highlighting the importance of self-reflection during the reasoning stage.

*From Code to Cases: Can LLM generate more comprehensive cases with the code?*

Given that the cases are closely related to code, a natural approach is to provide the model with a reference solution. By examining the structure of this solution, the model can better understand the underlying logic of the inputs and outputs for case generation. In our experiments, we prompt the model with correct human-written solutions and present the results in Figure 8. Our results show that **providing code solutions significantly improves the quality of the generated case sets**, particularly for challenging problems such as E and F. With access to validated human solutions that address various edge cases, the model can produce more comprehensive cases, covering a wide range of scenarios by leveraging this prior knowledge. Inspired by these findings, it becomes possible to generate accurate cases for problems that lack official ones for offline validation, which enables the creation of richer training data and supports more robust validation results in code training.

### 4.3 From Cases to X

*From Cases to Editorial: Is LLM also capable of judge cases?*

Following Section 4.2, we prompt the model to judge the correctness of test cases. The official truth cases are guaranteed to be correct, while the model-generated cases include both correct and incorrect examples. We exclude test cases whose total string length exceeds 200 and present the results in Figure 9. We observe that **the model is able to identify some incorrect cases**. Notably, for official test cases, the model achieves a judging accuracy of up to 90% for QwQ-32B, and this high accuracy remains consistent across different problem difficulties. In contrast, when evaluating self-generated cases, most misjudgments occur when the ground truth case fails but the model classifies it as correct. This suggests that **self-consistency still limits the model's ability to effectively judge test cases**, which aligns with the trends observed in Figure 6.

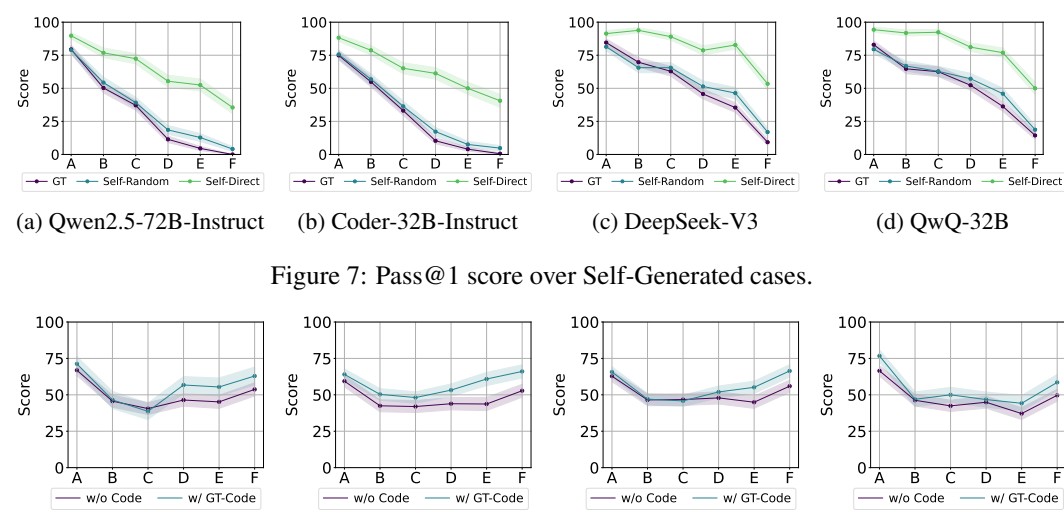

Figure 7: Pass@1 score over Self-Generated cases.

Figure 8: Case Score w.r.t. GT Solutions.

***From Cases to Code: Can self-generated solution pass all the self-generated cases?***

In this part, we explore another aspect of self-consistency: whether self-generated solutions can fully pass the test cases produced by the model itself. We construct two types of test cases: (1) *Self-Direct*, where the model directly generates both inputs and outputs; and (2) *Self-Random*, where the model generates random inputs, and the outputs are derived using validated solutions. As shown in Figure 7, **the pass rate on self-generated test cases is significantly higher than on the ground truth test cases**. This occurs because self-generated cases often lack comprehensive coverage, especially of edge conditions, and tend to remain within the bounds of the model's own understanding. Consequently, the accuracy on Self-Random and Self-Direct test cases can exceed that on ground truth test cases by up to 5% and 40%, respectively. Interestingly, for some problems that the model fails to solve entirely (e.g., Problem F), it can still achieve an "Accepted" result when evaluated on its own test cases. However, the pass rate on self-generated test cases is not 100% across all problems, indicating that **these test cases can still reveal errors in the self-generated solutions to some extent**. Therefore, using self-generated test cases to verify solution correctness remains a viable approach and offers opportunities for self-reflection and iterative self-improvement.

## 5 RELATED WORKS

**Coding Models.** Code generation has been fully influenced by LLMs, starting with Codex (Chen et al., 2021a), which powers GitHub Copilot and excels in converting natural language prompts into functional code. AlphaCode (Li et al., 2022) leverage an encoder-decoder architecture to solve complex algorithmic problems on platforms like Codeforces. Among open-source models, StarCoder (Lozhkov et al., 2024) stands out for its community accessibility and robust performance. Qwen2.5-Coder (Hui et al., 2024) achieves impressive results, rivaling GPT-4o, while DeepSeek-R1 (Guo et al., 2025) employs reinforcement learning to enhance reasoning and coding capabilities. General-purpose models like GPT-4 (Achiam et al., 2023) and o1 (OpenAI, 2025a) demonstrate exceptional code generation, highlighting the versatility and rapid evolution of LLMs in this domain.

**Coding Evaluation Benchmark.** Evaluating code generation models relies on robust benchmarks that primarily judge functional correctness. HumanEval (Chen et al., 2021a) serves as a foundational benchmark with 164 Python problems measure the ability of generating correct solutions. MBPP (Austin et al., 2021a) complements HumanEval by providing approximately 1,000 crowd-sourced Python tasks aimed at entry-level programmers. EvalPlus (Liu et al., 2023) further enhances evaluation rigor by expanding HumanEval and MBPP with extensive test cases. LiveCodeBench (Jain et al., 2024) introduces a dynamic, contamination-free evaluation with over 880 problems from platforms such as LeetCode and Codeforces, covering diverse tasks including code repair and test output

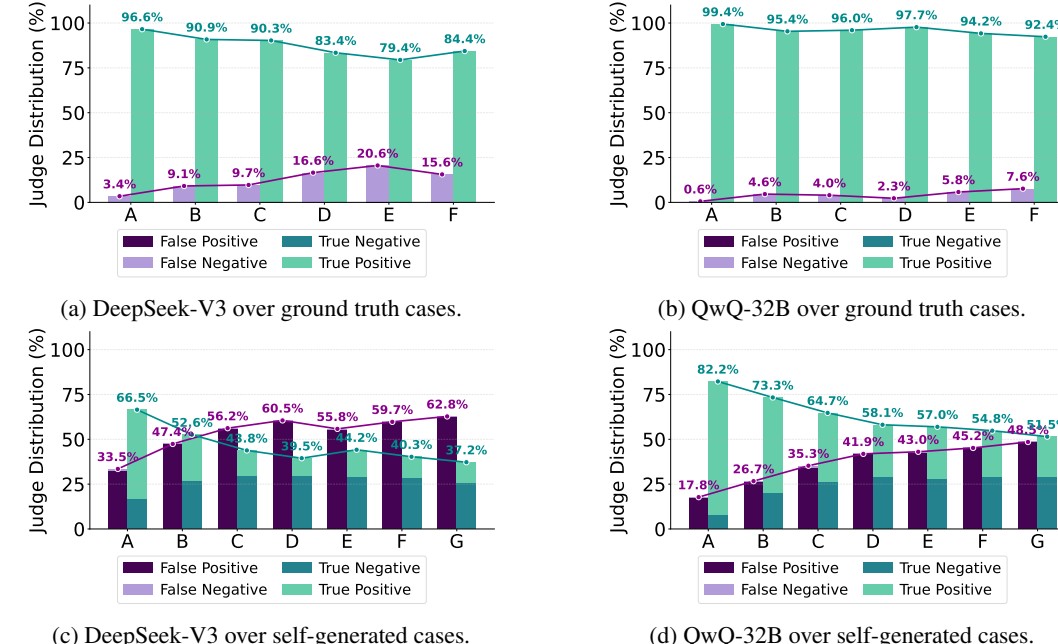

(a) DeepSeek-V3 over ground truth cases.

(b) QwQ-32B over ground truth cases.

(c) DeepSeek-V3 over self-generated cases.

(d) QwQ-32B over self-generated cases.

Figure 9: **Judge accuracy across cases.** LLM can correctly identify erroneous cases but tends to mistakenly regard its own generated cases as correct.

prediction. While these benchmarks collectively offer a comprehensive evaluation, they mainly focus on coding solutions and do not consider editorial analysis or test case generation.

## 6 DISCUSSION AND CONCLUSION

**Self-consistency exists within model cognition.** Analyzing the prediction across different dimensions, we observe that self-consistency exists within the model cognition. For example, leveraging the self-generated editorial does not lead to significant improvements in solutions. Besides, the roll-out solutions can easily pass the self-generated test cases, resulting in a kind of reward hacking with respect to test cases. These observations indicate the self-consistency within LLM cognitive stage due to limitations of training data, resulting in distribution shift with human submissions.

**Inconsistency across various dimensions may facilitate self-improvement.** LLM demonstrates different performance across the dimensions of the Coding Triangle and these differences can be further harnessed for self-improvement, such as using its own judgment to distinguish correct solutions, or leveraging a correct code to improve edge-case generation. This approach indicates that the development of self-improvement can be realized through iteratively align these dimensions, gradually reducing error correlations and develop LLMs with more powerful coding ability.

**Model mixture enhance diversity and robustness.** Recognizing the existence of self-consistency, we find that the combination of different models can significantly reduce bias while improving both robustness and diversity. For example, solutions generated by different models can help identify a wider range of boundary cases, and the combined test cases from multiple models are effective at detecting potential errors. These results demonstrate that leveraging model mixtures mitigates the biases and diverse outputs from different models contribute to better robustness.

**Limitation and Conclusion.** We systematically investigate the coding ability of LLMs through the lens of the coding triangle, i.e., editorial, code, and cases. There are still some limitations as we does not fully explore all possible interactions between these three dimensions. Overall, our study reveal the self-consistency and self-inconsistency inside LLMs, point out the importance of bridging the gap between model cognition and human expertise, and provide a potential direction of aligning and mutually reinforcing the three dimensions to achieve more reliable and generalizable coding models.

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

# APPENDIX

## A   DATASET AND LICENSE

The evaluating problem dataset is collected from AtCoder (https://atcoder.jp). We only collect publicly available content, including visible editorials, sample codes, and test cases published on the website. All collected materials are strictly used for research and evaluation purposes only, specifically to assess the performance of candidate models. No part of the collected dataset is used for training any model and we fully respect the content ownership and terms of use of AtCoder. We follow LiveCodeBench and abide by Fair Use §107: "the fair use of a copyrighted work, including such use by ... scholarship, or research, is not an infringement of copyright", where fair use is determined by "the purpose and character of the use, including whether such use is of a commercial nature or is for nonprofit educational purposes" and "the effect of the use upon the potential market for or value of the copyrighted work."

## B   LLM USAGE

In the preparation of this manuscript, we made limited use of LLMs as a general-purpose writing assistant. Specifically, the LLMs are employed to polish wording, improve sentence fluency, and adjust grammatical structure for clarity and readability. At no point did the LLMs contribute to research ideation, the formulation of hypotheses, methodological design, execution of experiments, or interpretation of results. Their role was strictly confined to surface-level language refinement, comparable to the functions of a grammar-checking or style-editing tool. All intellectual contributions, including the conception of ideas, development of approaches, and analysis of findings, are entirely the work of the authors.

## C   MORE EXPERIMENTS

In this section, we provide more experiments as additional results. We mainly focus on model mixture and the interaction across different models.

### C.1   DOES LLM BENEFIT FROM THE EDITORIAL FROM OTHER MODELS?

We further investigate whether editorials generated by other models can benefit code generation. To this end, we evaluate the performance of Coder-32B-Instruct when provided with editorials from Qwen2.5-72B-Instruct, DeepSeek-V3, and QwQ-32B, as shown in Figure 10. The results indicate that using editorials as prompts can serve as a form of knowledge distillation, enhancing the performance of the student model. With access to relatively accurate editorials from reasoning-oriented models, Coder-32B-Instruct, which excels at code implementation, can achieve performance comparable to that obtained with ground truth editorials, demonstrating the practicality of leveraging logical knowledge embedded in reasoning models to improve the code generation capabilities of coding models.

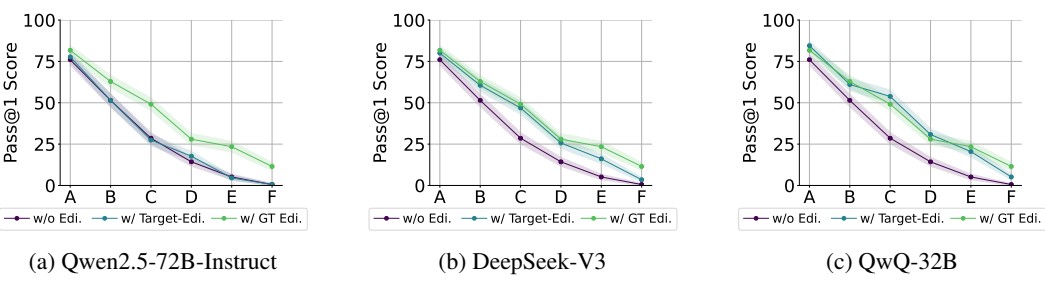

(a) Qwen2.5-72B-Instruct          (b) DeepSeek-V3          (c) QwQ-32B

Figure 10: Pass@1 score of Coder-32B-Instruct with editorials from other models.

## C.2 CAN LLM RECOGNIZE MISTAKES FROM OTHER MODELS?

We further conduct experiments to investigate whether LLMs can identify mistakes made by other models, aiming to explore the judging capability of LLMs. Specifically, we utilize DeepSeek-V3 to determine whether the solutions generated by other models contain errors. As shown in Figure 11, the model is also capable of identifying mistakes in the solution, and the accuracy all exhibits a decreasing-then-increasing trend.

For relatively weaker models (such as coder-32B and Qwen-72B), which lack the ability to solve difficult problems, DeepSeek-V3 can easily identify the errors in their solutions to such challenging problems, which is reflected by True Negatives constituting the majority of accurate judgments. Moreover, in the cases where DeepSeek-V3 judges its own solutions and those of QWQ-32B, we observe that DeepSeek-V3 shows a stronger tendency to judge its own solutions as correct, revealing a form of self-consistency in the model's self-perception.

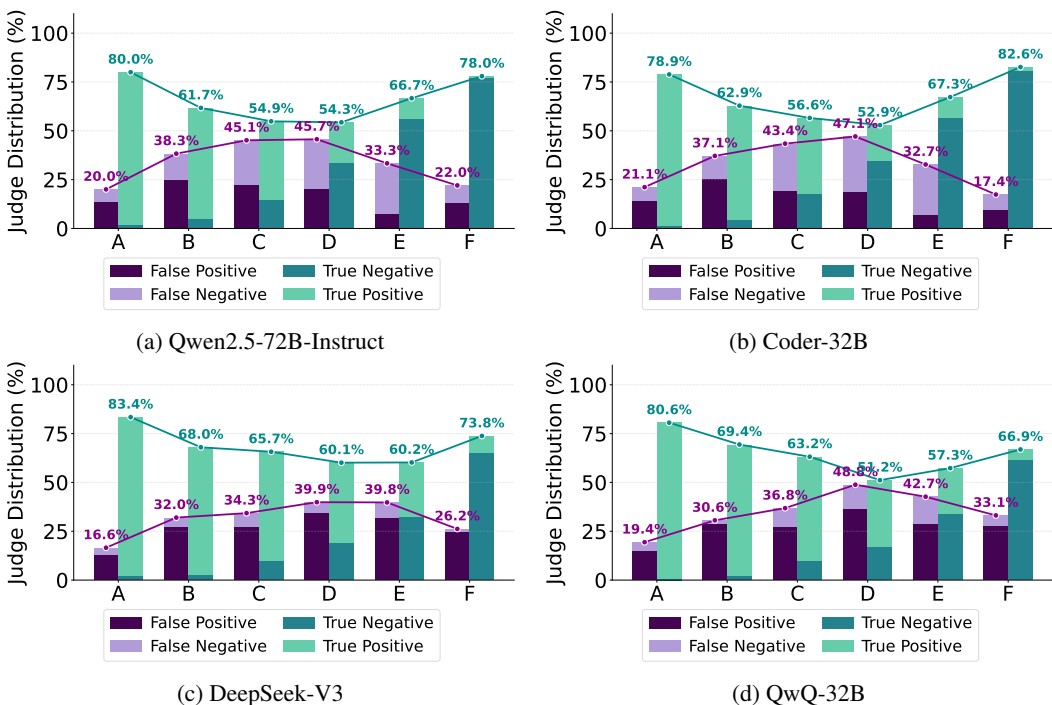

Figure 11: Judge distribution of DeepSeek-V3 over solutions from different models.

## C.3 HOW DOES CASE GENERATION BENEFIT FROM MODEL MIXTURE?

To further validate the effectiveness of model mixture, we merge the cases generated by different models and calculate the accuracy of these case sets. The results are presented in Figure 12. We observe that the model mixture brings significant improvements in case score performance, as different models exhibit different biases and the diversity introduced by these models in the case generation task directly contributes to the improvement in scores.

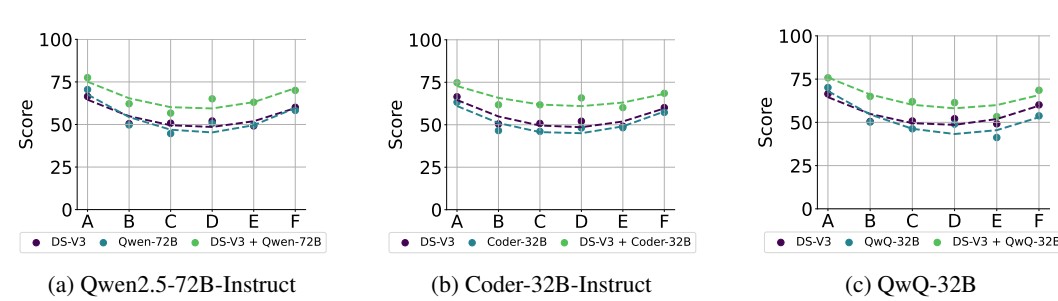

Figure 12: Case score of model mixture between DeepSeek-V3 and other models.

---

**Editorial Generation Prompt**

[[ Instruction ]]
You are a code expert, and your goal is to develop a editorial analysis for a given coding problem. You will analyze the problem, explain the approach, including any necessary constraints and mathematical formulas, to help the reader understand how to solve it. Your explanation should not include the final code but should guide the reader to implement it themselves. Please indicate the time complexity at the end of the analysis.

[[ Constraints ]]
Your editorial analysis must satisfy the following:
- Clearly explain the approach to solve the problem.
- Include any necessary constraints from the problem statement.
- Use mathematical formulas where applicable to clarify the solution.
- Do not provide the final code, only the editorial.
- Indicate the time complexity of the solution at the end.
- Ensure that the editorial addresses all aspects of the problem as stated.

[[ Example ]]
[Editorial]
The leaves of $T$ are leaves of initial stars, and those vertices distant by at most two from it belong to the same star. Thus, the following algorithm is possible.
- Choose a leaf of $T$.
- Count the number of vertices distant by at most two from it (including itself). If there are $x$ of them, they form a level-$(x-1)$ star.
- Remove the counted vertices and adjacent edges from $T$.
After you repeat this until no vertices remain in $T$, you obtain the answer. With an appropriate implementation, it works in a total of $O(N)$ time, but the implementation is a bit complicated. We describe a simpler implementation with some observations.
The following lemma holds:
- In an original star, let us call the non-leaf vertex the center. In $T$, the distance between centers is always a multiple of 3, and that between a center and a leaf is always a non-multiple of 3.
This can be shown by induction. With this lemma, we can come up with the following algorithm:
- Choose a vertex from $T$ that was the center of an original star. You can do so by choosing the vertex adjacent to a leaf.
- Find the shortest distance of each vertex from the chosen vertex.
- For each vertex whose shortest distance is a multiple of 3, add the degree of that vertex to $L$.
[Time Complexity] $O(N)$

[[ Output Format ]]
Your response should be formatted as follows and should not include any additional information:
[Think] Your thinking about the reasoning process in the mind.
[Editorial] Your final editoral of the problem.
[Time Complexity] The time complexity of your solution.

[[ Problem Begin ]]
{problem}
[[ Problem End ]]

---

756
757
758
759
760
761
762
763
764
765
766
767
768
769
770
771
772
773
774
775
776
777
778
779
780
781
782
783
784
785
786
787
788
789
790
791
792
793
794
795
796
797
798
799
800
801
802
803
804
805
806
807
808
809

## Solution Generation Prompt

[[ Instruction ]]
You are an expert C++ programmer. Your goal is to generate a complete, correct C++ program for a given coding problem. The program should handle all edge cases, follow best practices, and be efficient where necessary. Enclose your program within C++ code delimiters as shown below.
[[ Constraints ]]
- Generate a fully functional C++ solution that compiles and passes all tests.
- The code should be standalone and not rely on external libraries beyond what's standard in C++, unless specified in the problem.
- Adhere strictly to the problem's input and output formats.
- Ensure the code is clean, well-indented, and includes comments to explain complex logic.
- Include all necessary headers and use the standard namespace.
- Wrap your code in triple backticks with C++ annotation.
[[ Example ]]
#include <bits/stdc++.h>
using namespace std;
int main() {
// sample solution
return 0;
}
[[ Output Format ]]
Your response should be formatted as follows and should not include any additional information:
[Analysis] Your analysis of the problem.
[Code] Your C++ code in a code block.
[[Problem begin]]
{problem}
[[Problem end]]

810
811
812
813
814
815
816
817
818
819
820
821
822
823
824
825
826
827
828
829
830
831
832
833
834
835
836
837
838
839
840
841
842
843
844
845
846
847
848
849
850
851
852
853
854
855
856
857
858
859
860
861
862
863

## Case Generation Prompt

[[ Instruction ]]
You are an expert Python competitive programmer and your goal is to construct input-generators for testing programming contest problems. You will write relevant generators and finally implement a 'construct_inputs' function that returns a list of 50 diverse inputs sampled from those generators. Remember to strictly follow the instructions and constraints present in the problem statement.

[[ Constraints ]]
Your input-generators and 'construct_inputs' must satisfy all of the following:
- **Deterministic framework**: the code may call randomness internally, but the overall scheme and parameter ranges must be hard-coded (no external configuration or user prompts).
- **Coverage**: include edge-case ranges (smallest/largest sizes, boundary weight values), typical scenarios, and stress scenarios near the problem's limits.
- **Validity**: generated inputs must always respect the problem's stated input format and numeric bounds (e.g. $1 \leq N \leq N_{max}, weight_{min} \leq weight_i \leq weight_{max}$, etc.).
- **Reproducibility**: allow for seeding if needed (e.g. accept a seed parameter), but default behavior needs no external input.
- **Diversity**: return a list containing at least three tiers of size/scale (e.g. small, medium, large) and within each tier cover multiple parameter combinations.
- **Clarity**: each testcase's 'input' string must be parseable by the contestant's code.

[[ Example ]]

```python
import numpy as np
def random_input_generator(weight_min, weight_max, size_min, size_max, seed=None):
    if seed is not None:
        np.random.seed(seed)
    n = np.random.randint(size_min, size_max+1)
    weights = np.random.randint(weight_min, weight_max+1, size=n).tolist()
    k = np.random.randint(1, n+1)
    return { 'input': ' '.join(map(str, weights)) + ' ' + str(k) + '\\n' }

def edge_case_generator(case_id):
    cases = [
        # 0: smallest size, smallest weight
        { 'input': '1 1\\n' },
        # 1: smallest size, largest weight
        { 'input': '1 1000000000\\n' },
        ...
        # 9: mixed boundary in medium
        { 'input': '1000 ' + ' '.join(['1']*499 + ['1000000']*500) + ' 250000\\n' },
    ]
    return cases[case_id]

def construct_inputs():
    inputs_list = []
    # 10 edge cases
    for i in range(10):
        inputs_list.append(edge_case_generator(i))
    # small tier
    for i in range(10, 20):
        inputs_list.append(random_input_generator(1, 10**3, 1, 10, seed=i))
    # medium tier
    for i in range(20, 35):
        inputs_list.append(random_input_generator(1, 10**6, 1, 10**3, seed=i))
    # large tier
    for i in range(35, 50):
        inputs_list.append(random_input_generator(1, 10**9, 1, 10**5, seed=i+100))

    return inputs_list
```

[[ Output Format ]]
Your response should be formatted as follows and should not include any additional information:
[Analysis] Your analysis of the problem.
[Code] Your Python scripts in a code block.
[[ Problem Begin ]]
{problem}
[[ Problem End ]]

**Editorial Judge Prompt**

[[ Instruction ]]
You are a code expert and judge. Your goal is to evaluate a candidate's editorial for a given coding problem, using the official editorial as a reference. First, extract the time complexities from both the official editorial and the candidate's editorial. If the candidate's time complexity is asymptotically worse than the official one, assign a score of 0. Otherwise, analyze whether the candidate's solution is logically correct and solves the problem as required. Note that the candidate's approach may differ from the official one, but it should still be a valid solution to the problem. Finally, assign a binary score: 1 if the solution is correct and has an acceptable time complexity, otherwise 0.
[[ Constraints ]]
Your judgment must satisfy the following:
- Read and understand the problem statement in full.
- Use the official editorial as a reference to verify the correctness of the candidate's approach, but allow for different valid solutions.
- Assign a score of 1 if the candidate's solution is logically correct and has an acceptable time complexity, otherwise assign 0.
[[ Output Format ]]
Your response must follow exactly this format without any extra information:
[Analysis] Your analysis of the problem, the two editorials, and the correctness of the candidate's solution.
[Score] Your final judge score.
[[ Problem Begin ]]
{problem}
[[ Problem End ]]
[[ Official Editorial Begin ]]
{gt editorial}
[[ Official Editorial End ]]
[[ Candidate Editorial Begin ]]
{editorial}
[[ Candidate Editorial End ]]

**Solution Judge Prompt**

[[ Instruction ]]
You are a programming competition judge. Your task is to analyze a submitted solution for a specified problem and determine its correctness. You should focus on logical correctness, coverage of all edge cases, and any implementation flaws that would cause test failures.
[[ Constraints ]]
- Provide a detailed analysis of potential logical errors or omissions.
- Indicate whether the solution passes all test cases or fails some.
- Do not execute code; base your judgment on static reasoning.
- Assign a score of 1 if the candidate's solution is logically correct and has an acceptable time complexity, otherwise assign 0.
[[ Example ]]
[Analysis] The solution attempts binary search but has an off-by-one error in the termination condition (line 8). This causes incorrect results when the target is at array boundaries.
[Score] 0
[[ Output Format ]]
Your response should be formatted as follows and should not include any additional information:
[Analysis] Your analysis of the problem and the solution.
[Score] Your 0/1 score of the solution.
[[ Problem begin ]]
{problem}
[[ Problem end ]]
[[ Solution begin ]]
{solution}
[[ Solution end ]]

## Case Judge Prompt

[[ Instruction ]]
You are a programming competition judge. Your task is to determine whether a given test case's input and output match according to the problem's specification. Focus solely on whether the provided output is the correct result for the provided input under the problem logic.
[[ Constraints ]]
* Check that the "Case Output" is exactly what the problem would produce for the given "Case Input."
* Identify any discrepancies, incorrect results, or mismatches.
* Do not assess solution code—only compare input versus output.
* Do not execute code; base your judgment on logical reasoning and the problem statement.
* Assign a score of 1 if the candidate's case is logically correct, otherwise assign 0.
[[ Example ]]
[Analysis] For input '3 5 2', the problem asks for the sum of the first two numbers. The expected result is '8', but the provided output is '15', so they do not match.
[Score] 0
[[ Output Format ]]
Your response must follow this exact format, with no additional text:
[Analysis] <your detailed analysis of input-output matching>
[Score] <0 or 1>
[[ Problem begin ]]
{problem}
[[ Problem end ]]
[[ Case Input Begin]]
{case input}
[[ Case Input end ]]
[[ Case Output Begin ]]
{case output}
[[ Case Output end ]]

