# OpenReview forum: "Coding Triangle: How Does Large Language Model Understand Code?"
_ICLR.cc/2026/Conference — Submitted to ICLR 2026_

### Official Review · Reviewer_XSWv · 2025-10-30

**Soundness:** 3
**Presentation:** 3
**Contribution:** 3
**Rating:** 4
**Confidence:** 4

**Summary:**

This paper addresses an important topic in the discussion of understanding how LLMs understand code. It introduces the framework of a **Coding Triangle** that covers 3 different ways models address and understand coding problems. While each individual component is not novel, the framework is a novel way of combining the various components. U

**Strengths:**

- Framework of the Coding Triangle is novel and original
- The writing is clear and easy to follow for the majority of the paper and the results are sensible as well.
- In particular the insights on self-consistency are insightful.
- The idea and framework could be very impactful

**Weaknesses:**

- There are not enough models evaluated. Most notably, the paper doesn't cover core models like GPT, Sonnet, Gemini, etc.
- Evaluating on AtCoder seems relatively limited. Are there any other benchmarks where you can do this analysis also?
- While the paper brings in the idea of a coding triangle, the paper doesn't thoroughly address how each component connects to one another. The main connection discussed by the paper is how editorials affect code generation, but the reverse or how cases affect code or how cases affect editorials are not discussed. The paper should try to cover all combinations to be thorough.
- An obvious extension would be seeing if the model could revise its solution based on its test cases and if that improves performance. I might have missed this, but don't see it in the paper.
- Minor, but needs more detail on what AtCoder is, the problem count, why A-F is different, etc.

**Questions:**

Last two points in weaknesses are my questions

---

### Official Review · Reviewer_F4GB · 2025-10-31

**Soundness:** 3
**Presentation:** 3
**Contribution:** 3
**Rating:** 6
**Confidence:** 3

**Summary:**

This paper proposes "Coding Triangle," a three-dimensional framework for evaluating LLMs' programming capabilities through editorial analysis, code implementation, and test case generation. The authors conduct comprehensive experiments on competitive programming problems, revealing interesting phenomena of self-consistency and self-inconsistency in model behaviors. The work addresses an important gap in current code evaluation benchmarks and provides valuable insights into model cognition.

**Strengths:**

The three-dimensional evaluation framework is innovative and addresses limitations of existing benchmarks

The analysis of self-consistency and self-inconsistency reveals important characteristics of model cognition

The discovery that model mixtures enhance diversity and robustness is practically valuable

Comprehensive experiments across multiple model types and problem difficulties

**Weaknesses:**

The evaluation is limited to competitive programming problems; generalization to real-world coding scenarios needs verification

The "self-consistency" and "self-inconsistency" concepts could be more precisely defined and quantified

Limited analysis of why reasoning models still exhibit self-inconsistency despite extended reasoning capabilities

No discussion about the computational cost of implementing the full Coding Triangle evaluation

**Questions:**

1. How would your framework perform on more practical coding tasks like software maintenance or code review, beyond algorithmic problems?

2. Have you considered applying your findings to improve model training, perhaps by explicitly addressing the identified self-consistency issues?

3. The hidden state in recurrent models maintains some memory - could similar mechanisms help LLMs maintain consistency across coding tasks?

4. What's the computational overhead of implementing all three dimensions of your evaluation framework compared to traditional benchmarks?

---

### Official Review · Reviewer_8kHe · 2025-11-01

**Soundness:** 2
**Presentation:** 1
**Contribution:** 1
**Rating:** 2
**Confidence:** 5

**Summary:**

The paper evaluates three dimensions of code generation: (1) Code, (2) Editorial, and (3) Case. It proposes no new methods or datasets. All experiments use Qwen series models and Deepseek V3. The paper reports several interesting observations: (1) LLMs show self-consistency between editorial and code dimensions, (2) code generation capabilities are similar across different LLMs but distinct from human submissions, (3) human submissions are more diverse than LLMs on code and cases, (4) human editorials are more helpful than LLM-generated editorials for code generation, (5) LLMs can recognize their own mistakes, (6) LLMs can generate more comprehensive test cases than they can solve.

**Strengths:**

No obvious grammar flaw in the paper.

**Weaknesses:**

1. Figure 1, the teaser is difficult to follow, I can’t understand the relationship between green, blue, and orange arrows and blocks. And which dimensions are self-consistent or not self consistent cannot easily tell from the figure.
2. The evaluation models: QWQ, Qwen coder and Qwen instruct are basically from the same company, my concern is I think their would be some similarity in pretrain data, a more diverse model to be used would make the observations in the paper seems more reasonable.
3. As an evaluation paper, there is no qualitative analysis in the paper, only outcome based analysis make this paper shallow in the depth of analysis.
4. Again without in-depth analysis and explaination, I cannot understand why human editorial is better than human generated editorials on code and cases generation.
5. In the section: From Code to Cases: Can LLM generate more comprehensive cases with the code? How to make sure the generated cases is inherently correct? How complex the cases are compared to the original seed cases.
6. According to the authors, the LLMs can find their own mistakes, I am confused in what extent they can find mistakes, in what extent they can’t? And why models can find mistakes but cannot solve the problem directly? would test-time scaling help? would self-refinement such as ReAct help?
7. To what extent the model cannot find its own mistakes? is there any qualitative anlysis on that?
8. All experiments are down via Pass@1, would the conclusions in the paper be different if use test-time scaling to evaluate?
9. The experiment setting is unclear, for example the temperature of LLMs to decode, thus the similarity experiment in Figure 3 is not that accurate from my perspective, temperature could affect the diversity by a lot. And I am not sure why human generated code is more diverse than LLMs? is it because the code are submitted by different people, and the author use an outcome based analysis, I think is not enough to say they are more diverse.

From my humble perspective, the paper is not polished enough, lots of experiment settings and qualitative analysis are missing. The evaluation method is not surprising or make sense enough. I think the paper need a major revision, thus I would rate a maximally score of 2 to the paper.

**Questions:**

See Weakness

---

### Official Review · Reviewer_J9TQ · 2025-11-01

**Soundness:** 2
**Presentation:** 3
**Contribution:** 2
**Rating:** 4
**Confidence:** 4

**Summary:**

This paper investigates the underlying code understanding capabilities of Large Language Models (LLMs), arguing that current benchmarks focused on functional correctness are insufficient. The authors propose a new, three-dimensional evaluation framework called the "Coding Triangle," which assesses models across: (1) Editorial (natural language problem analysis), (2) Code (algorithm implementation), and (3) Cases (test case generation).

Through experiments on AtCoder problems, the authors analyze the "self-consistency" and "self-inconsistency" of LLM cognition. The primary contributions are:

The identification of high "self-consistency" in LLMs, which leads to a lack of solution diversity and a significant "distribution shift" when compared to human-generated solutions .

The finding that this self-consistency limits exploration (e.g., models' self-generated editorials do not improve their coding performance , and their code easily passes their own flawed test cases ).

The observation that "self-inconsistency" also exists (e.g., models can sometimes identify their own failed solutions ) and that "model mixture" can improve robustness by increasing diversity.

**Strengths:**

* Framework: The primary strength is the proposal of the "Coding Triangle" (Editorial, Code, Cases). This is a novel, intuitive, and significant contribution. It provides a multi-dimensional, interpretable framework that moves beyond simple functional correctness to probe an LLM's analytical and validation capabilities.



* Insight: The paper clearly identifies and provides evidence for "self-consistency" and "distribution shift". The finding that LLM-generated solutions are highly similar (high cosine similarity) and lack the diversity of human solutions is an important one for the community.



* Connecting Self-Consistency to Exploration: This work provides an explanation for the success of exploration-based sampling techniques. By showing that models are often trapped in their own narrow "cognition" , the paper highlights that breaking this self-consistency (e.g., via "model mixture" ) is a good way to improve robustness.



* Holistic Analysis: The analysis of the interactions between the triangle's vertices (the "edges" ) is a high-quality contribution.

**Weaknesses:**

* Methodological Opacity (Critical Weakness): As detailed in the "Soundness" section, the paper is missing the most crucial experimental details. The authors analyze solution diversity and self-consistency without specifying the decoding parameters (temperature, top-p, etc.) or the number of samples (k) used for the diversity analysis in Figure 3. These parameters are not minor details; they are the central variables that control the exploration and diversity the paper claims to measure. This omission makes the core experimental results non-reproducible and unverifiable.



* Limited Exploration of Diversity Enhancement: The paper correctly identifies "model mixture" as a way to enhance diversity. However, this is a very expensive (inference-time) and somewhat obvious solution. The paper would be much stronger if it explored other, single-model methods for increasing exploration as a counterpoint. For example, how do the diversity metrics (Fig 3) and robustness scores change if one simply increases the sampling temperature for a single model? Does that achieve the same gains as the model mixture? The discussion feels incomplete by only proposing model mixture as the solution to the self-consistency problem.



* Ambiguous "Cases" Metric: The metric for the "Cases" dimension, S_{case}, is convoluted and its reliability is unclear. It is defined by the ability to identify "incorrect human submissions" (\mathcal{H}_{i}^{wrong}). How large is it? How was it curated? A model might get a high S_{case} score by generating a few simple cases that catch common errors, which does not fully capture the "depth of understanding in terms of validation criteria" that the authors claim to measure.

**Questions:**

Most of the questions are mentioned in the weakness part. Additional one:

The paper proposes "model mixture" to increase diversity. Have the authors experimented with single-model techniques for increasing exploration, such as increasing the sampling temperature or using nucleus sampling with a high p? How do these simpler methods compare to the expensive "model mixture" approach in terms of increasing robustness and the "Unique Set Size"?

---

### Meta-Review · Area_Chair_ioqF · 2026-01-08

**Summary:**

The authors proposed Coding Triangle, a new evaluation framework to evaluate LLMs by 3 aspects: code editing, code implementation, and test case generation. The authors found both self-consistency and self-inconsistency in models.

**Reviewer Concerns:**

- There are missing details in experimental setup e.g. temperature, the number of samples k, etc. These parameters are critical to ensure the results are reproducible and verifiable
- The experimental setup is limited to QwQ, Qwen Coder, and Qwen Instruct which are from the same model family.
- There are critical presentation and writing issues. For instance, the definition of “cases” metric is ambiguous and not well defined. Figure 1 is not clear and hard to tell the relationship between arrows and blocks. The concepts of self-consistency and self-inconsistency are not defined well.

**Reviewer Scores:**

The authors did not respond during the rebuttal. Therefore, reviewers would not change their scores.

---

### Decision · Program_Chairs · 2026-01-26

Reject